# In Vitro Susceptibility of Clinical and Carrier Strains of *Staphylococcus aureus* to STAFAL^®^ Phage Preparation

**DOI:** 10.3390/ijms252312885

**Published:** 2024-11-29

**Authors:** Marek Straka, Zuzana Hubenáková, Lucia Janošíková, Aneta Bugalová, Andrej Minich, Martin Wawruch, Adriana Liptáková, Hana Drahovská, Lívia Slobodníková

**Affiliations:** 1Institute of Microbiology, Faculty of Medicine, Comenius University in Bratislava, 81108 Bratislava, Slovakia; marek.straka@fmed.uniba.sk (M.S.); lucia.janosikova@fmed.uniba.sk (L.J.); livia.slobodnikova@fmed.uniba.sk (L.S.); 2Institute of Biology, Faculty of Medicine, Slovak Medical University, 83101 Bratislava, Slovakia; zuzana.hubenakova@szu.sk; 3Comenius University Science Park, Ilkovičova 8, 84104 Bratislava, Slovakia; lichvarikova.aneta@gmail.com (A.B.); hana.drahovska@uniba.sk (H.D.); 4Medirex Group Academy, Novozámocká 1, 94905 Nitra, Slovakia; andrej.ado.minich@gmail.com; 5Institute of Pharmacology and Clinical Pharmacology, Faculty of Medicine, Comenius University in Bratislava, 81108 Bratislava, Slovakia; martin.wawruch@gmail.com; 6Department of Molecular Biology, Faculty of Natural Sciences, Comenius University in Bratislava, Ilkovičova 6, 84104 Bratislava, Slovakia

**Keywords:** *Staphylococcus aureus*, MRSA, STAFAL^®^, nasal carriage, skin and soft tissue infections, bloodstream infections

## Abstract

The treatment of infections caused by *Staphylococcus aureus* is currently complicated by the increasing number of strains resistant to antimicrobial agents. One promising way to solve this problem is phage therapy. Due to the lack of data on the effectiveness and safety of phage preparations, STAFAL^®^ is the only registered phage preparation for the treatment of infectious diseases in the Slovak Republic and the entire European Union. The aim of this work was to determine the effectiveness of the STAFAL^®^ phage preparation against *S. aureus* strains of different origins with variable sensitivity to antimicrobial substances and with different genetic backgrounds. For this purpose, 111 carrier strains, 35 clinical isolates from bloodstream infections, and 46 strains from skin and soft tissue infections were analysed. The effectiveness of STAFAL^®^ was determined by the plaque forming method. STAFAL^®^ was effective against 74.0% of the strains tested. Susceptibility to this phage preparation was significantly higher in strains resistant to methicillin (MRSA), erythromycin and clindamycin (*p* < 0.05). The high efficiency of the STAFAL^®^ preparation was confirmed against spa types t003, t024 and t032, typical of the hospital environment. The in vitro results indicate high therapeutic potential of the STAFAL^®^ antistaphylococcal phage preparation, especially against MRSA strains.

## 1. Introduction

*Staphylococcus aureus* is an opportunistic pathogen that commonly colonises the nasal mucosa and human skin [1,2]. However, colonisation by *S. aureus* may be the basis for the development of infection [3]. This bacterium is capable of causing a wide range of infections, from skin and soft tissue infections to systemic infections, such as pneumonia, osteomyelitis, sepsis or endocarditis; thus, it is a leading cause of infection-related mortality worldwide [1,2,4]. The development of infection is facilitated by its set of virulence factors [2,5].

The treatment as well as decolonisation of *S. aureus* from body sites to prevent infectious events is complicated by the increasing number of strains resistant to beta-lactam antibiotics, designated as MRSA strains, that are circulating in the population. These strains are also often able to acquire resistance to other groups of antimicrobial agents (ATBs)—macrolides, tetracyclines, aminoglycosides, chloramphenicol and quinolones [3,6]. Moreover, the use of vancomycin as the final line in the treatment of MRSA infection has led to decreased susceptibility and treatment failure in some cases [7,8]. Other significant limitation in the treatment of *S. aureus* by ATBs include the risk of renal failure, allergy, drug–drug interactions and intolerable side effects, particularly in immunocompromised patients in the hospital environment [9,10,11]. For these reasons, it is necessary to search for new strategies that would be effective in the treatment of infections caused by resistant strains of microorganisms—in this particular case, by *S. aureus* [12,13].

Treatment by bacteriophages may be an alternative to classic antibiotic therapy, or its suitable supplement [14]. Bacteriophages (shortly phages) are viruses capable of specifically lysing bacterial cells. They are the most widespread form of organism in the world, present everywhere where bacteria are found [15].

Phage therapy has several advantages over conventional antibiotic therapy. Firstly, due to the specificity of the phage, it has a minimal impact on the physiological microbiota of the host. After the elimination of host bacteria, phages naturally diminish [12,13]. Phage therapy is considered safe for the human body, since phages are not toxic to or mutagenic in mammalian cells [13,16]. Another advantage is that resistance of bacteria to phages is developed 10 times slower than the resistance to antibiotics [17,18]. An indisputable advantage is the lower financial burden compared with the development and production of new antibiotics. The disadvantages of phage therapy include the possibility of neutralising antibody production by the patient’s body, which can reduce the effectiveness of bacteriophages [19]. Another disadvantage is the potential spread of genes encoding virulence or antibiotic resistance. The spread of these genes can be eliminated by whole-genome sequencing and the subsequent selection of phages that do not contain such genes [20]. Despite the limitations mentioned, treatment by bacteriophages may be the last possible therapeutic option in some cases [21,22,23,24,25]. In addition to the therapy of infections, phages can also be used to decolonise *S. aureus* carriers [19].

Bacteriophages were discovered as early as 1915 (and separately in 1917) and were used for the treatment of infectious diseases, such as staphylococcal skin infections, cholera or dysentery [26,27,28]. However, after the discovery and introduction of antibiotics into clinical practice, phage therapy was abandoned in most parts of the world. The production of phage preparations for the treatment of infectious diseases was continued in Eastern Europe in countries such as Georgia or Russia. However, these phage preparations are not recognised by Western regulatory agencies. Nowadays, due to the emergence of multidrug-resistant (MDR) strains, there is a restoration of efforts to produce safe and effective phage preparations also in other countries, such as France, Belgium, the United Kingdom, Australia, China, India or the United States [17,29,30,31,32,33]. Actually, phage therapy in Poland is administered to patients in the Phage Therapy Unit as a personalised experimental therapy [30]. In Slovakia, the antistaphylococcal phage preparation STAFAL^®^ is currently the only phage preparation registered in a member state of the European Union [32,34].

The STAFAL^®^ phage preparation includes lytic polyvalent bacteriophages of the family *Herelleviridae*, genus *Kayvirus* [35,36]. STAFAL^®^ is indicated for the elimination of *S. aureus* from infectious foci and from possible reservoirs of infection (mainly from the upper respiratory tract—nose, nasopharynx and paranasal sinuses—and secondarily also from the gastrointestinal and uropoietic systems). It is also a drug for the treatment of purulent wounds, chronic skin and subcutaneous tissue infections and infections affecting deep-seated soft tissue and for the prevention of septic state development [34].

The aim of this work was to determine the in vitro efficacy of the phage preparation STAFAL^®^ against clinical and even carrier *S. aureus* strains with variable susceptibility to antimicrobial compounds and variable genetic backgrounds in Slovakia, where the phage preparation is already registered.

## 2. Results

### 2.1. Antimicrobial Susceptibility of Isolated Staphylococcus aureus Strains

The summary antimicrobial susceptibility of *S. aureus* strains analysed in this study is shown in Table 1. Thirty-three MRSA strains belonging especially to the group of clinical strains (29 isolates—87.9%) were included. A total of 3 carrier strains and 24 clinical isolates were classified as MDR. One hundred and ten carrier strains (99.1%) were susceptible to local antimicrobial agents mupirocin, neomycin, bacitracin and fusidic acid.

The susceptibility of MRSA strains to the other tested ATBs was statistically lower compared with the susceptibility of MSSA strains. These differences were detected in the susceptibility to erythromycin (36.6% vs. 67.3%), clindamycin (39.4% vs. 73.0%) and ciprofloxacin (21.2% vs. 91.2%) (Table 2).

### 2.2. Susceptibility to STAFAL^®^

The overall efficacy of the STAFAL^®^ phage preparation was confirmed in 142 *S. aureus* strains (74.0%). Slightly higher susceptibility to the phage preparation was demonstrated in clinical strains compared with carrier strains (73.9% vs. 74.1%), but these differences were not statistically significant (*p* = 0.985). The highest susceptibility was assessed in strains isolated from bloodstream infections (80.0%), followed by carrier strains (73.9%) and strains from skin and soft tissue infections (69.6%). No statistically significant differences were detected either (*p* = 0.571) (Figure 1).

The susceptibility of the tested bacterial strains to STAFAL^®^ was different in the strains susceptible to ATBs and the strains resistant to ATBs. A statistically significant difference was found in the phage susceptibility of MRSA strains (31 phage-susceptible strains (93.9%)) and MSSA strains (111 phage-susceptible strains (69.8%)). Two other statistically significant difference were found in the phage susceptibility of erythromycin-resistant (60 phage-susceptible strains (82.2%)) and erythromycin-susceptible strains (82 phage-susceptible strains (68.9%)) and in the susceptibility of clindamycin-resistant (53 phage-susceptible strains (82.8%)) and clindamycin-susceptible strains (89 phage-susceptible strains (69.5%)) (Figure 2).

The efficacy of STAFAL^®^ was confirmed in 23 MDR strains (85.2%), and the susceptibility of non-MDR strains was detected in 119 cases (72.1%). A non-significant difference was detected in the phage susceptibility of MDR and non-MDR strains to STAFAL^®^ (*p* = 0.235).

The STAFAL^®^ phage preparation was effective against strains belonging to 63 of the detected 97 spa types (64.9%). Five spa types included strains with variable susceptibility (5.2%). Strains belonging to the other 29 spa types were resistant to STAFAL^®^ (29.9%) (Table 3).

## 3. Discussion

Nowadays, the complications associated with the treatment of infections caused by MDR strains are emerging. Phage therapy constitutes an alternative way of treating infectious diseases. However, only a few phage preparations are currently registered, since there is still a lack of studies confirming the effectiveness and safety of this type of products [32,38,39,40,41]. The only phage preparation registered within the European Union is STAFAL^®^. This phage preparation is indicated for topical application, especially for the treatment of purulent infections of the skin and soft tissues caused by *S. aureus* [32,34].

The basic prerequisite for the use of phage preparations in the therapy of infections in patients is their ability to lyse bacterial cells in vitro [42,43]. Therefore, 192 *S. aureus* strains of Slovak origin were selected to evaluate the therapeutic potential of the STAFAL^®^ phage preparation. This collection included clinical strains isolated from infected patients, but also carrier strains from the nasal mucosa of healthy students of medicine, since they participate in healthcare facilities during their internships and nasal carriage contributes significantly to the spread of staphylococcal infections in vulnerable individuals [44,45]. The other aim of this study was to test the effect of STAFAL^®^ on MRSA strains, since they are often associated with other types of resistance and these strains are often the reason for complicating therapy [46,47,48].

The antibacterial activity of the STAFAL^®^ phage preparation has already been proven in studies by Dvořáčková et al. [49,50]. In the first study, the effectiveness of this preparation was assessed against clinical strains of MRSA and MSSA, and higher susceptibility of MRSA was detected (more than 99%). MSSA strains were susceptible in 87.9% of cases [49]. The high effectiveness of the antistaphylococcal preparation against clinical strains in planktonic form and in biofilm was determined in the second study. The strains analysed in this work were included in spa types t003, t024, t032 and t056 [50].

In this study, the analysis of these clinical and carrier strains of *S. aureus* revealed 74.0% efficacy of the STAFAL^®^ phage preparation. Similar effectiveness of phage preparations from Russia, Bakteriophag Stafilokokovyj (74.5%) and Sextaphag^®^ (73.4%), and Georgia, Staphylococcal bacteriophage (75.6%), was detected in this collection of strains in the previous study. The efficiency of the Georgian preparation Pyo-bacteriophage (80.2%) was slightly higher. These phage preparations and STAFAL^®^ had similar spectra of action in our strain collection. Most of the strains were susceptible to all these five phage preparations, including STAFAL^®^ (67.7%). On the other hand, 16.1% were resistant to all the preparations tested. Of all *S. aureus* strains, 83.9% were susceptible to at least one of the phage preparations tested [37].

The susceptibility of *S. aureus* strains to STAFAL^®^ was compared in terms of the origin of the strains, their susceptibility to ATBs, and their molecular genetic affiliation. Slightly higher susceptibility of clinical strains compared with carrier strains to STAFAL^®^ was detected. The highest efficacy of STAFAL^®^ was observed in the group of strains of bloodstream infection isolated in severe-to-life-threatening infections. Similar results were also detected for the other four phage cocktails [37].

The susceptibility of the tested strains to STAFAL^®^ varied in the strains with different susceptibility to ATBs. The tested phage preparation showed significantly higher activity against ATB-resistant strains (except tetracycline), especially against MRSA strains. Significantly higher phage susceptibility was obtained also for erythromycin and clindamycin-resistant strains. Similar results were also observed in this strain collection during the testing of the Russian and Georgian preparations [37]. Oechslin [51] stated that bacterial mutations that lead to resistance to phages often result in an increased survival cost for this bacterium. Additionally, the effect of two distinct inhibiting factors may reduce the risk of resistance development [52].

The efficacy of the STAFAL^®^ preparation has been confirmed against a genetically diverse range of *S. aureus* strains, including typical nosocomial spa types t003, t024 and t032 [53,54,55,56,57]. Strains within the same spa type had similar patterns of susceptibility to the phage preparation, and only a few exceptions were found. Similarly, it was described also for the previously tested preparations. In the case of three strains (affiliated with t362, t701 and t1333), their susceptibility to STAFAL^®^ was detected, while the other four phage preparations were ineffective. On the other hand, four different strains (affiliated with t008, t3884, t6943 and t12469) were susceptible to all the preparations tested, except for the STAFAL^®^ preparation. Six other strains (affiliated with t091) and two strains (affiliated with t2716) were susceptible to Pyo-bacteriophage, but STAFAL^®^ was not effective [37]. *S. aureus* strains with declared resistance to the STAFAL^®^ preparation can be used as indicator strains to expand the spectrum of this preparation. The resistance of the bacterial strains to the phage preparation may be a result of mutation in receptors for phage adsorption, acquiring genes encoding restriction enzymes or CRISPR-Cas system-cleaving phage nucleic acid, the blocking of virus progeny release, etc. [58,59].

The American Antibiotic Resistance Leadership Group and Health Improvement Scotland have recommended the consideration of phage therapy for difficult-to-treat bacterial infections [9,60]. Our results confirm that the use of the STAFAL^®^ phage preparation may be beneficial in the treatment of bacterial infections, primarily in patients infected or colonised by MRSA strains [17,38,61]. Moreover, it may be also a beneficial tool for the decolonisation of *S. aureus* from body sites [3] The successful application of the STAFAL^®^ preparation in patients with infections caused by *S. aureus* took place in the 1960s and 1970s in the Czech Republic [62]. More recent experience with the treatment of *ulcus cruris* in chronic venous insufficiency, vasculitis, diabetic legs or severe carbuncles by STAFAL^®^ in Slovakia was described in a study by Zelenková [63]. Here, the beneficial effect of STAFAL^®^ was also confirmed.

The efforts to treat infections caused by *S. aureus* by other phages or phage preparations with antistaphylococcal activity have already been described. Pyo-bacteriophage was administered by intravesical titration in a double-blind study to patients with urinary tract infection. The effect of the phage cocktail was comparable with the efficacy of ATBs and placebo, but it was well tolerated [64]. Ferry et al. [65] used a phage cocktail (including phages PP1493, PP1815 and PP1957) followed by antibiotics for the treatment of three patients with prosthetic knee infection with significant clinical improvement in all of them. In another study, a patient with knee and hip prosthetic joint infection by MRSA underwent the exchange of a new spacer and was treated by phage Sa WIQ0488ø1 and daptomycin. Finally, all bacteriological cultures were negative, and no evidence of recurrence was detected [66]. The intravenous administration of phage cocktail AB-SA01 was described in patients with bacteremia by Petrovic Fabijan et al. [67]. Patients were also treated by ATBs. No adverse effects were reported, and probably, a synergic effect between phages and ATBs was achieved.

Despite the successful experience with phages in vitro or even in clinical use, STAFAL^®^ is still the only phage preparation registered in a member state of the European Union and, in addition, is currently unavailable [32,34].

## 4. Materials and Methods

### 4.1. Characteristics of Bacterial Strains

In this study, 111 carrier strains and 81 clinical isolates of *S. aureus* characterised in a former study [37] were submitted to analysis. The carrier strains were obtained from nasal swabs of healthy volunteers—students of the Faculty of Medicine, Comenius University in Bratislava. The clinical isolates originated from the samples of patients of University Hospital Bratislava indicated for microbiological examination, where 35 isolates were from bloodstream and 46 from skin and soft tissue infections. Carrier strains were isolated from nasal swabs cultivated on BD Columbia Agar with 5% Sheep Blood (Becton Dickinson GmbH, Heidelberg, Germany). *S. aureus* strains were identified according to the cultivation properties and positive hyaluronidase test [68]. Clinical strains were identified within the routine microbiological diagnostics at University Hospital Bratislava [68]. Strains with uncertain identification were analysed by PCR according to Martineau et al. [69]. The antimicrobial susceptibility of carrier strains to erythromycin (15 µg), clindamycin (2 µg), tetracycline (30 µg), trimethoprim–sulfamethoxazole (co-trimoxazole) (2 µg + 23 µg) and ciprofloxacin (5 µg) (all Oxoid, Basingstoke, UK) was tested by the disc diffusion test according to EUCAST recommendations [70]. In this group of strains, the susceptibility to local ATBs, such as mupirocin (200 µg), bacitracin (200 µg), neomycin (200 µg) and fusidic acid (200 µg), was also assessed [70]. The screening of MRSA strains was conducted by the disc diffusion test by using cefoxitin (30 µg) [69], and confirmation in both carrier and clinical strains was conducted by the detection of the *mecA* gene by PCR according to Martineau et al. [71]. Strains resistant to at least one ATB from three or more ATB groups were classified as multidrug-resistant (MDR) [72].

The molecular characterisation of all *S. aureus* strains was performed by *spa* typing based on the sequence typing of the spa gene repeat region [73,74]. Bacterial genomic DNA was isolated from an overnight culture of the isolated strain in 10 mL of LB medium (BioLife, Milan, Italy) by using the Illustra TM bacteria genomicPrepMini Spin Kit according to the manufacturer’s instructions (GE Healthcare, Chicago, IL, USA). The amplification of the spa gene was carried out in 25 µL reactions by using primers spa-1113f (TAAAGACGATCCTTCGGTGAGC) and spa-1514r (CAGCAGTAGTGCCGTTTGCTT) (Merck KGaA, Darmstadt, Germany). The PCR reaction contained 2 µL of isolated DNA, 320 µM deoxynucleoside triphosphates (Solis bioDyne, Tartu, Estonia), 12 pmol of each primer, 10-fold concentrated DreamTaq green Buffer and 1.25 U of DreamTaq DNA polymerase (both ThermoFisher Scientific, Waltham, MA, USA). Thermal cycling reactions consisted of initial denaturation (2 min at 94 °C) followed by 35 cycles of denaturation (45 s at 94 °C), annealing (45 s at 66 °C) and extension (90 s at 72 °C), with final extension (10 min at 72 °C) in the thermocycler Biometra TAdvanced 96 (Analytik Jena, Jena, Germany). The presence of amplicons was detected by separation in 1% agarose gel. Subsequently, the PCR products were purified with the illustra TM ExoProStar TM kit (GE Healthcare, Chicago, IL, USA) according to the manufacturer’s instructions. Sequencing was performed by using the BigDye Terminator v3.1 Cycle Sequencing Kit (Applied Biosystems, Waltham, MA, USA) with an ABI PRISM 3130xl instrument (Applied Biosystems, Waltham, MA, USA). The obtained DNA sequences were analysed by using BIONUMERICS software version 7.5 (Applied Maths, Sint-Martens-Latem, Belgium). New spa types were uploaded to the Ridom SpaServer database [75].

### 4.2. Testing of Susceptibility of Staphylococcus aureus Strains to STAFAL^®^ Phage Preparation

*S. aureus* strains were subjected to the testing of susceptibility to the STAFAL^®^ phage preparation (Bohemia Pharmaceuticals, Brno, Czech Republic, currently AUMED, Praha, Czech Republic [76]). This preparation contains 10^7^ phage particles per mL that are preserved by thiomersal.

The susceptibility of bacterial strains to this preparation was determined by the plaque formation method as described in a previous study [37]. After overnight cultivation of *S. aureus* strains at 37 °C on BD Columbia Agar with 5% Sheep Blood (Becton Dickinson GmbH, Heidelberg, Germany) a standardised inoculum of bacterial strains (McFarland 0.5, corresponding to 1–5 × 108 CFU/mL) was prepared turbidimetrically, using Densitometer DEN-1 (Biosan, Riga, Latvia). Luria–Bertani agar (BioLife, Milan, Italy) plates were overlaid with 2 mL of standardised bacterial inoculum. The inoculum excess was sucked out by a pipette. After the inoculum had soaked into the agar, the phage preparation STAFAL^®^ was point-inoculated in 10 μL volumes of non-diluted phage preparation and preparation diluted in 1:10, 1:100 and 1:1000. Incubation lasted for 16 h at 37 °C. The activity of the phage preparations was determined by the detection of confluent bacterial lysis, semi-confluent plaques, or individual isolated plaques in the spot area of inoculated phage suspensions. Any of these reactions at any dilution were considered positive. The effect of thiomersal on bacteria was distinguished from the phage effect by testing different dilutions of the phage preparation and the preservative itself.

### 4.3. Statistical Analysis

The distribution of categorical variables between two groups was compared by using the Pearson χ^2^ test. Fisher’s exact test was used if the expected count was less than five in ≥20% of cells of the 2 × 2 contingency table. The Fisher–Freeman–Halton exact test was applied to compare the distribution of categorical variables among the three groups in 2 × 3 tables. All the statistical tests were performed at the level of statistical significance α = 0.05. The analyses were performed by using IBM SPSS statistical software for Windows, version 28 (IBM SPSS Inc., Armonk, NY, USA).

## 5. Conclusions

In the “post-antibiotic era”, phage therapy may constitute a helpful tool to manage the treatment of bacterial infections caused by MDR strains and thus decrease the rate of morbidity and mortality in patients. In this study, the high in vitro efficacy of the STAFAL^®^ phage preparation against strains isolated in Slovakia with various genetic background indicates its therapeutical potential, especially in the infections caused by MRSA strains. Due to several advantages that phages have over ATBs, STAFAL^®^ preparation may play an important role as an alternative or complementary treatment, especially in case of failure of ATB therapy. Moreover, there is a potential for use of this phage preparation as a decolonising agent for carriers.

## Figures and Tables

**Figure 1 ijms-25-12885-f001:**
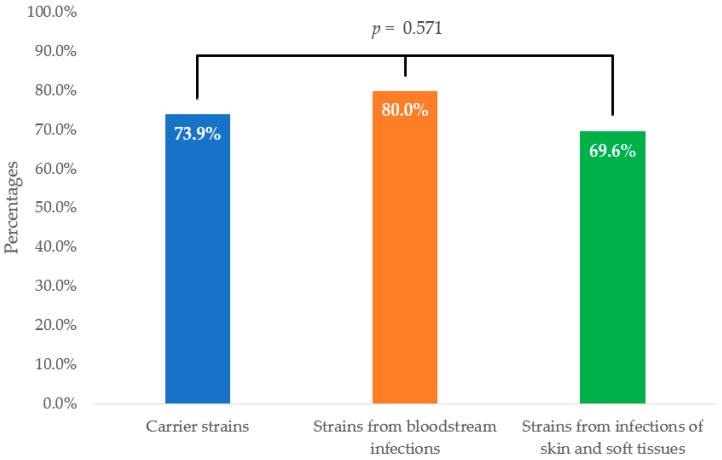
Comparison of efficacy of STAFAL^®^ against *Staphylococcus aureus* strains according to their origin. *p*—statistical significance among carrier strains, isolates from bloodstream infections and strains from skin and soft tissue infections according to Fisher–Freeman–Halton exact test.

**Figure 2 ijms-25-12885-f002:**
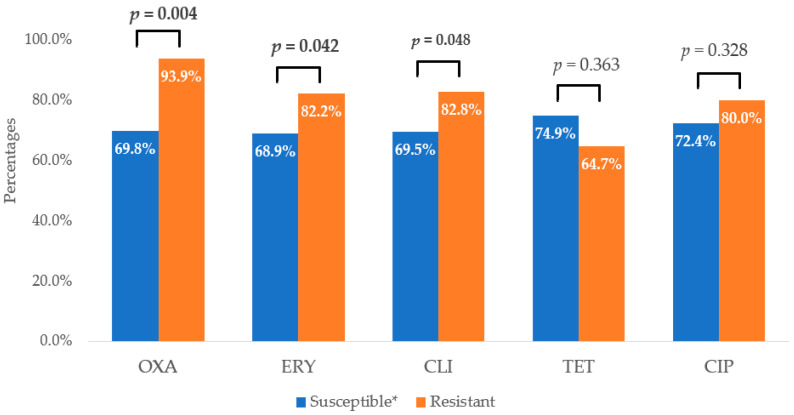
Comparison of efficacy of STAFAL^®^ against ATB-susceptible and ATB-resistant strains of *Staphylococcus aureus*. OXA—oxacillin; ERY—erythromycin; CLI—clindamycin; TET—tetracycline; CIP—ciprofloxacin; *p*—statistical significance between susceptible and resistant strains to various antimicrobial compounds according to χ^2^-test; in the case of statistical significance (*p* < 0.05), the values are expressed in bold; * in the case of ciprofloxacin—susceptible, increased exposure.

**Table 1 ijms-25-12885-t001:** Susceptibility of *Staphylococcus aureus* strains to antimicrobial compounds [37].

ATB	Carrier Strains *n* = 111 (100.0)	Strains from Bloodstream Infections *n* = 35 (100.0)	Strains of Infections of Skin and Soft Tissues *n* = 46 (100.0)	Total *n* = 192 (100.0)	*p*-Value
OXA					**<0.001**
S	107 (96.4)	27 (77.1)	25 (54.3)	159 (82.8)	
R	4 (3.6)	8 (22.9)	21 (45.7)	33 (17.2)	
ERY					**<0.001**
S	79 (71.2)	22 (62.9)	18 (39.1)	119 (62.0)	
R	32 (28.8)	13 (37.1)	28 (60.9)	73 (38.0)	
CLI					**0.004**
S	81 (73.0)	26 (74.3)	21 (45.7)	128 (67.6)	
R	30 (27.0)	9 (25.7)	25 (54.3)	64 (33.3)	
TET					0.443
S	102 (91.9)	30 (85.6)	43 (93.5)	175 (91.1)	
R	9 (8.1)	5 (14.3)	3 (6.5)	17 (8.9)	
COT					-
S (I)	111 (100.0)	35 (100.0)	46 (100.0)	192 (100.0)	
CIP					**<0.001**
I	108 (97.3)	23 (65.7)	21 (45.7)	152 (79.2)	
R	3 (2.7)	12 (34.3)	25 (54.3)	40 (20.8)	
MDR					**<0.001**
No	108 (97.3)	29 (82.9)	28 (60.9)	165 (85.9)	
Yes	3 (2.7)	6 (17.1)	18 (39.1)	27 (14.1)	

The values of categorical variables represent frequency, and the percentages are provided in parentheses (% of n). ATB—antimicrobial compound; OXA—oxacillin; ERY—erythromycin; CLI—clindamycin; TET—tetracycline; COT—co-trimoxazole; CIP—ciprofloxacin; S—susceptible to ATBs; R—resistant to ATBs; I—susceptible, increased exposure; MDR—multidrug resistant; *p*—statistical significance among groups of carrier strains, strains from bloodstream infections and strains from skin and soft tissue infections according to the Fisher–Freeman–Halton exact test; in the case of statistical significance (*p* < 0.05), the values are expressed in bold.

**Table 2 ijms-25-12885-t002:** Comparison of the susceptibility profiles of methicillin-resistant (MRSA) strains and methicillin-susceptible (MSSA) strains of *Staphylococcus aureus* to other antimicrobial compounds.

ATB	MRSA *n* = 33 (100.0)	MSSA *n* = 159 (100.0)	Total *n* = 192 (100.0)	*p*-Value
ERY				**<0.001**
S	12 (36.6)	107 (67.3)	119 (62.0)	
R	21 (63.6)	52 (32.7)	73 (38.0)	
CLI				**<0.001**
S	13 (39.4)	115 (72.3)	128 (66.7)	
R	20 (60.6)	44 (27.7)	64 (33.3)	
TET				1.000 *
S	30 (90.9)	145 (91.2)	175 (91.1)	
R	3 (9.1)	14 (8.8)	17 (8.9)	
COT				-
S (I)	33 (100.0)	159 (100.0)	192 (100.0)	-
CIP				**<0.001**
I	7 (21.2)	145 (91.2)	152 (79.2)	
R	26 (78.8)	14 (8.8)	40 (20.8)	

The values of categorical variables represent frequency, and the percentages are provided in parentheses (% of n). ATB—antimicrobial compound; ERY—erythromycin; CLI—clindamycin; TET—tetracycline; COT—co-trimoxazole; CIP—ciprofloxacin; S—susceptible to ATBs; R—resistant to ATBs; I—susceptible, increased exposure; *p*—statistical significance between the MRSA and MSSA groups by χ^2^-test; * statistical significance according to Fisher’s exact test; in the case of statistical significance (*p* < 0.05), the values are expressed in bold.

**Table 3 ijms-25-12885-t003:** Susceptibility of *Staphylococcus aureus* strains of various molecular genetic affiliations to STAFAL^®^.

Susceptibility to STAFAL^®^	Spa Types
Susceptible *n* = 63 (64.9)	t003, t007, t010, t012, t014, t018, t024, t026, t036, t045, t056, t084, t085, t122, t148, t156, t160, t169, t189, t209, t223, t267, t279, t284, t342, t346, t360, t362, t435, t449, t491, t493, t648, t701, t706, t718, t760, t774, t922, t937, t1148, t1200, t1265, t1309, t1333, t1491, t1509, t2119, t2124, t2374, t3382, t3732, t4032, t4559, t4688, t5534, t16302, t16466, t18619, t18623, t18626, t18627 and t18629
Resistant *n* = 29 (29.9)	t004, t015, t050, t065, t091, t216, t289, t571, t688, t715, t728, t1040, t1255, t1646, t2248, t2642, t2716, t2932, t3625, t3884, t4545, t6608, t6943, t7157, t12469, t12588, t18621, t18625 and t18628
Variable susceptibility *n* = 5 (5.2)	t002, t008, t032, t179 and t1451

Percentages are provided in parentheses (% of *n*).

## Data Availability

Data are contained within the article.

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
