# Peer review of "In Vitro Susceptibility of Clinical and Carrier Strains of Staphylococcus aureus to STAFAL® Phage Preparation"

_ijms, 2024, doi:10.3390/ijms252312885_

Round 1

Reviewer 1 Report

Comments and Suggestions for Authors

The study appears to lack novelty, particularly in terms of novel insights, advancements in methodology, clinical applications, or a deeper understanding of phage-bacteria interactions beyond what is already well-established in the literature. The authors have previously published similar studies using different phage preparations. It is unclear why this study specifically focuses on the STAFAL® phage preparation, which is already registered and has presumably undergone prior efficacy and safety evaluations to gain approval for therapeutic use in Slovakia and the European Union.

Other Comments:

  • Subsections must be numbered.
  • Data provided in lines 140-153 are repeated in Figure 2.
  • Decimal points should be used for separation in figures.
  • Provide Y-axis titles for both figures.
  • The methodology section lacks clarity. For example, details are missing on how isolates were prepared for the experiments and the CFU/ml concentrations used.
  • The origin of chemicals and manufacturers of instruments used in the study should be provided.
  • Lines 273-275 state that, "In carrier strains, susceptibility to local ATBs, such as mupirocin (200 µg), bacitracin (200 µg), neomycin (200 µg), and fusidic acid (200 µg) was also assessed," yet no data regarding this assessment is provided.
  • The study’s conclusion appears to validate and reinforce existing data on the potential of phage therapy, particularly for MDR bacterial infections, rather than presenting novel insights.
  • Although nasal swabs were collected from healthy volunteers to obtain carrier strains, the study still requires informed consent, ethical approval, and adherence to confidentiality and data protection protocols to ensure ethical conduct.
  • Latin names in the reference list must be italicized.
  • Verify the journal requirements to determine whether the full or abbreviated journal titles should be used.

Reviewer 2 Report

Comments and Suggestions for Authors

The publication describes important studies in a practical aspect, however, in my opinion, in terms of content, they fit better into other journals, e.g. medical (e.g. Pharmaceuticals) or microbiological (e.g. Microorganisms), because the authors do not delve into the molecular mechanism of STAFAL® action. There is no detailed answer as to why some strains are resistant to phages. Is it perhaps the result of biofilm formation? What is the effect of phages on the formation of biofilm by S. aureus? Do the phages studied have the ability to penetrate biofilm? In the text of publication, there is also no characterization of the lytic activity of phages.

The methodology lacks a description of the conditions for culturing both S. aureus strains (media, culture age, OD, etc.) and phages. There are also no conditions for purifying phages, even patent references. There is no source for STAFAL® and no information on how the preparation was revived. How was the amount of phages administered checked so that this amount was comparable between experiments? How was the viability of the administered preparation checked?

Correct identification of strains is crucial for drawing correct conclusions. However, references 71, 72 (line 280) do not contain a detailed description of the procedure for identifying S. aureus strains. Since this is crucial for the quality of the publication, a detailed description of the method used should be included in the text of the publication.

Reference 54 refers to the microdilution method and not the disc diffusion test. This publication also lacks details of the method used.

Line 32 it should be „in vitro” instead of „in vitro”.

Line 79 it should be „[26,27,28]” instead of „(26,27,28)”.

Line 279 it should be „spa” instead of „spa”, because it is a gene.

Latin names of microorganism strains in references should also be written in italics.

Round 2

Reviewer 1 Report

Comments and Suggestions for Authors

I agree with the improvements of the manuscript.

Just noticed comma instead of period in Fig.1.

Reviewer 2 Report

Comments and Suggestions for Authors

The authors have carefully revised the manuscript and in my opinion, in its current form it is suitable for publication.